# Coal-Fired Boiler Flue Gas Desulfurization System Based on Slurry Waste Heat Recovery in Severe Cold Areas

**DOI:** 10.3390/membranes12010047

**Published:** 2021-12-30

**Authors:** Chenghu Zhang, Dezhi Zou, Xinpeng Huang, Weijun Lu

**Affiliations:** 1School of Architecture, Harbin Institute of Technology, Harbin 150001, China; chenghu.zhang@163.com (C.Z.); 18845787916@163.com (W.L.); 2Inner Mongolia Key Laboratory of Green Building, College of Architecture, Inner Mongolia University of Technology, Huhhot 010051, China

**Keywords:** slurry temperature, heat and mass transfer, wet flue gas desulfurization, desulfurization efficiency, recovery of waste heat

## Abstract

To reduce operating costs on the basis of ensuring the desulfurization efficiency in a wet flue gas desulfurization system, a theoretical model was put forward, and a calculation method was set up. Correlations between reaction zone height, flue gas inlet temperature, slurry inlet temperature, gas–liquid ratio and desulfurization efficiency were found. Based on the heat and mass transfer model of the spray tower, the integrated system of desulfurization tower and open slurry pool and the flue gas desulfurization-waste heat recovery system were established. Additionally, the effect of outdoor wind speed, heat dissipation area and ambient temperature on the slurry equilibrium temperature in the integrated system were analyzed. The results show the slurry equilibrium temperature of the desulfurization system is negatively correlated with outdoor wind speed and heat dissipation area, and positively related to ambient temperature. The slurry temperature is the main factor that affects the performance of the wet flue gas desulfurization system. Finally, based on the Harbin heating group Hua Hui hotspot energy-saving reconstruction project, a case analysis was conducted, which proves the flue gas desulfurization-waste heat recovery system is profitable, energy saving and a suitable investment project.

## 1. Introduction

High-temperature exhaust not only pollutes the environment but also causes huge energy waste. Therefore, to the creation of methods to effectively deal with pollutants in high-temperature exhaust gas and recycle waste heat energy, is one of the urgent energy problems to be solved in China. The higher the flue gas emission temperature, the greater the energy saving space [1]. Generally speaking, the exhaust temperature of coal-fired boilers is between 160 °C and 250 °C. The flue gas temperature of coal-fired boilers in most parts of China is much higher than the design value, which is generally 20–50 °C higher. According to the empirical value, every time the exhaust gas temperature rises by 10 °C, the coal consumption will increase from 1.2% to 2.4%, and the heat loss of flue gas exhaust at high temperatures will increase from 0.6% to 1.0% [2].

At the same time, in view of the existing phenomenon of coal-fired flue gas desulfurization in China, there is an unreasonable desulfurization slurry temperature. When the temperature of the desulfurization slurry is too high, the equilibrium pressure of the sulfur dioxide in the tower will increase, resulting in low desulfurization efficiency. The excessively high flue gas temperature may also cause the service life of the brittle material in the tower to decrease. Other effects include the increase of water evaporation of the desulfurization slurry, the increase of the desulfurization water replenishment rate and the increase of net flue gas humidity, which may cause corrosion to equipment [3,4].

Wet magnesium desulfurization has been developed in China for more than ten years, and its reliability, structural and economic operation feasibility, in addition to the feasibility of the desulfurization of by-product utilization have also been widely recognized [5]. As early as the 1990s, Matsui [6] and other studies proved that alkaline solutions such as magnesium hydroxide slurry have excellent absorption of sulfur dioxide and other various acid gases in flue gas and exhaust gas, which makes magnesium desulfurization technology widely used. At the same time, various scholars have used different methods to conduct various studies to improve the efficiency of desulfurization. Berman [7] et al. studied the absorption of sulfur dioxide from flue gas by magnesium hydroxide slurry in a convection jet tower, and the desulfurization rate was more than 94%. Hayakawa [8] tried to oxidize magnesium sulfite in the slurry of wet magnesium desulfurization to magnesium sulfate by air and then added lime to promote the regeneration of magnesium hydroxide. The results of this research prove that the method can reduce the amount of absorbent and greatly reduce the cost of desulfurization. However, research on using waste heat recovery technology to extract heat from the desulfurization slurry is not reported.

At present, domestic and foreign scholars have proposed a variety of design models to solve the heat and mass transfer of wet desulfurization. Studies have shown that the commonly used models for solving gas absorption in the liquid phase are rate-based and equilibrium-based models [9]. However, for the equilibrium-based model, after introducing the reaction kinetics, the equilibrium-based model is greatly expanded. However, since the heat transfer and mass transfer are not taken into account in the model, the accuracy of the model is reduced. On the contrary, it also reduces the research on multi-component gas–liquid mass transfer [10,11]. Hikita et al. [12] conducted a detailed study on the absorption process of sulfur dioxide by a sodium-alkali solution and constructed an SO_2_ absorption rate model based on the double membrane theory. However, as the research continues, it was found that it was difficult to solve the system of nonlinear differential equations in the model. After extensive research, they finally chose the approximate analytical solution, which is a better method to solve the system of nonlinear differential equations and the results are not very different from the origional ones. Baokui Chen et al. [13] put forward a theoretical model to predict the slurry temperature in a wet flue gas desulfurization system. They used a one-dimension coupled model using droplet motion, heat and mass transfer and the pressure loss of liquid–gas phases. Zongliang Qiao et al. [14] used a data mining method to find the best operation conditions for a wet flue gas desulfurization system. Their conclusions can provide operation guidance for a 600 MW generation unit. Mingchen Qin et al. [15] developed a numerical model and applied it to predict dual-loop spray tower 200 MW coal-fired unit efficiency, and the desulfurization system resistance loss agreed well with the test data. It can be seen that the current design model of the heat and mass transfer of wet desulfurization is complex, so it is of high research value to simplify the model and design a set of simple calculation programs.

Considering the operating cost of the desulfurization process, a novel system was proposed in the present work to increase the desulfurization efficiency and utilize the waste heat of the slurry. Taking the operating characteristics of the system into account, a new mathematical model was developed and the influence of the heat and mass transfer process, unsteady operation characteristics and thermal and economics characteristics of the system were analyzed. Consequently, the results can serve as a guide for alternative applications in power plants.

## 2. Calculation Models for the Wet Flue Gas Desulfurization System

### 2.1. Assumptions and Simplifications for Models

In the process of setting up the model, it is necessary to neglect some subordinate factors. Some reasonable assumptions and simplifications are made as follows:The slurry droplet is a regular sphere with uniform particle size and the same physical parameters at the same position. The mutual friction and collision between the droplets are not considered in the process of motion.The countercurrent process of flue gas and droplets is sufficient, uniform and stable.The droplets and flue gas move at a constant speed in the process of movement.Ignore the interference of the existing chemical reactions on heat and mass transfer.

### 2.2. Model for Heat and Mass Transfer

The slurry is introduced through injection nozzles and converted into large amounts of droplets in the atomization process. Flue gas flows in from the side of the tower to the top of the tower in a counter-current mode with the limestone slurry droplets. The slurry droplets evaporate into a counter-current flue gas stream. Heat and mass transfer occur on the surface of the droplets. In the mass exchange process, when other factors are neglected, the mass of the slurry reduction should be equal to the change in the moisture content between the slurry droplets and the flue gas. In the process of mass exchange, when other factors are neglected, as the water in the slurry enters the flue gas, the reduced mass of the slurry should be equal to the change of moisture content between the slurry droplets and the flue gas. Considering the diffusion of flue gas and slurry, the mass change can be expressed as:(1)$$dm_{d}=k_{gd}\cdot A_{gd}\cdot \left(x_{g}-x_{d}\right)\cdot \frac{dX}{u_{d}}$$

Therefore, the liquid mass conservation equation can be written as:(2)$$\frac{4\pi }{3}\cdot \rho_{d}\cdot dR_{d}{}^{3}=k_{gd}\cdot A_{gd}\cdot \left(x_{g}-x_{d}\right)\cdot \frac{dX}{u_{d}}$$

The heat of solution *Q_r_* is mainly determined in two parts of heat: one is convective heat transfer *Q_d_*, and the other is vaporization latent heat *Q_q_*.

Differential form is written as: (3)$$dQ_{d}=h_{gd}\left(t_{gi}-t_{di}\right)A_{i}dX$$
(4)$$dQ_{q}=r_{wtg}\cdot dm_{d}$$

Reconfiguring Equations (1)–(4), the liquid energy conservation equation can be written as: (5)$$c_{p,d}\cdot \rho_{d}\cdot \frac{4\pi }{3}{\left(\frac{D_{d}}{2}\right)}^{3}\frac{dt_{d}}{dX}=h_{gd}\cdot \pi \cdot D_{d}^{2}\left(t_{g}-t_{d}\right)+\frac{k_{gd}\cdot \pi \cdot D_{d}^{2}\left(x_{g}-x_{p}\right)r_{wtg}}{u_{d}}$$

When high-temperature flue gas encounters low-temperature desulfurization slurry, a mass transfer occurs on the surface of the contact between slurry and flue gas. The saturated moisture content of flue gas at slurry temperature is compared with that of flue gas itself to get the direction of mass transfer. The mass flow *M_x_* of moisture entering the micro-element from the slurry should be equal to the mass flow *M_x_*_+__Δ*x*_ of moisture flowing out of the micro-element. The gas-phase humidity change equation can be written as:(6)$$u_{g}\cdot \rho_{g}\cdot A_{0}\cdot \frac{dx_{g}}{dX}=\frac{N\cdot k_{gd}\cdot A_{gd}\cdot \left(x_{g}-x_{d}\right)}{u_{d}}$$

The derivation of the flue gas energy conservation equation is similar to that of the slurry energy conservation equation [16]. The heat release of flue gas is equal to the heat gain of slurry, so the gas energy conservation equation can be expressed as: (7)$$u_{g}\cdot A_{0}\cdot \rho_{g}\cdot di_{wa}=u_{d}\cdot c_{p,d}\cdot \rho_{d}\cdot \frac{4\pi }{3}{\left(\frac{D_{d}}{2}\right)}^{3}dt_{d}$$
(8)$$i_{wa}=c_{p,g}t_{g}+(r_{wtg}+c_{p,w}t_{g})x_{g}$$

According to mass conservation, it is known that the derivation of the mass conservation equation of S entering the micro-element is similar to the derivation of the flue gas humidity differential equation [17]. The corresponding expression can be written as: (9)$$\frac{d(u_{g}c_{so_{2}})}{dX}=k_{so_{2}}A_{gd}(c_{so_{2}}-c^{*}{}_{so_{2}})$$
(10)$$\eta =\frac{c_{so_{2}}-c^{*}{}_{so_{2}}}{c_{so_{2}}}$$

The pressure of the flue gas in the desulfurization tower concerns one atmosphere, and the solubility of sulfur dioxide in the slurry is very small, which meets the condition of the dilute solution. Therefore, the equilibrium concentration of gas–liquid absorption obeys Henry’s law.
(11)$$c^{*}{}_{so_{2}}=\frac{m_{so_{2}}}{H_{x}Rt_{g}}$$
(12)$$dm_{so_{2}{}_{i}}=\frac{m_{g_{i}}c_{so_{2}{}_{i}}-m_{g_{i-1}}c_{so_{2}{}_{i-1}}}{\rho_{g}m_{d_{i}}}$$

Henry’s constant [18] can be written as: (13)$$H_{x}=\frac{A}{\rho }\exp(-\frac{E}{T})$$

### 2.3. Calculation of Heat and Mass Transfer Model

A numerical solution is used to analyze the heat and mass transfer model. Before the numerical calculation, the effective reaction zone of the desulfurization tower is treated by grid micronization. It is regarded as a whole, and the inlet section to the outlet section of the reaction zone is divided into n equal parts. The height of each part is set to a small unit Δ*x*, the corresponding tower cross-sectional area of each small unit is Δ*A*_0_ and the segmented node diagram is shown in Figure 1. In height, Δ*x_i_* and Δ*x_i_*_+1_ are two adjacent compute nodes, and the initial parameters of the next position are calculated by the previous node.

Assuming that the temperature and moisture content of the flue gas out of the desulfurization tower are known, the temperature and moisture content of the flue gas needed in the second section can be obtained by taking it into the first section of the reaction zone. Then, the temperature and moisture content of the flue gas needed in the third section can be obtained by taking the last solution into the second section of the reaction zone to solve the problem. The temperature and moisture content of the flue gas from the desulfurization tower leaving the reaction zone can be obtained by repeating the above steps. These parameters are known before the calculation, and the calculated parameters are compared with those given. If they are not equal, then adjust them until they are equal. Finally, when the two values are equal, the assumed temperature and moisture content of the flue gas out of the desulfurization tower are correct. The flowchart is shown in Figure 2.

The working media and structural parameters of the desulfurization tower involved in solving the model are shown in Table 1.

### 2.4. Integrated System of Desulfurization Tower and Open Slurry Pool

Figure 3 show the treatment of desulfurization slurry by the desulfurization tower. The heat exchange of the desulfurization slurry consists of five parts: heating an amount of slurry in the desulfurization scrubber *Q*_1_, heat dissipation *Q*_2_ caused by water evaporation in the aeration tank, sedimentation tanks and circulation tanks, heat dissipation *Q*_3_ from convective heat transfer, heat loss *Q*_4_ from the aeration tank, sedimentation tank and the surface, pool bottom, pool wall, pipeline and equipment of the circulation pool and heat loss *Q*_5_ caused by sewage discharge and water replenishment.

When the heating capacity is greater than the heat dissipation capacity, the slurry temperature will continue to rise during the cycle. After the slurry temperature increases, the heating temperature difference of the slurry in the desulfurization tower is reduced, and the heating amount is decreased. At the same time, the heat dissipation of the aeration tank, sedimentation tank and the circulation tank and the convective heat transfer increase, so the slurry temperature rises slowly in the later stage. During the cycle, the amount of heat and heat dissipation will gradually approach a steady-state equilibrium. The model of the integrated system is based on this principle.

According to the energy conservation, the following formula can be obtained:(14)$$Q_{1}-Q_{s}=M_{d}c_{p,d}\frac{d\bar{t_{d}}}{d\tau }$$

The heating capacity of the desulfurization tower is mainly caused by the contact heat transfer between the slurry and the flue gas in the tower. The heat obtained from the slurry belongs to the total heat transfer, which can be obtained by convection according to the relationship between the convection heat transfer and the total heat transfer. The formula is as follows:(15)$$Q_{1}=\omega \cdot h_{gd}\cdot A_{gd}\cdot \left(\bar{t_{g}}-\bar{t_{d}}\right)$$

Figure 4a show the relationship between convective heat transfer and the total heat varies with the reaction zone height of the desulfurization tower. The equation obtained from the data can be written as:(16)$$\omega_{1}=0.0222X+1$$

From Figure 4b, when flue gas temperature is constant, and the slurry inlet temperature is changed, the relationship between convective heat transfer and total heat transfer can be expressed as:(17)$$\omega_{2}=1.714-0.023\cdot t_{d}-2.64\times 10^{-5}\cdot t^{2}{}_{d}$$

Reconfiguring Equations (16) and (17), the ratio of convective heat transfer to total heat transfer is shown in Table 2. Take the average of *ω*_1_, and we get $\bar{\omega }$_1_ = 1.0444; take the average of *ω*_2_, and we get $\bar{\omega }$_2_ = 0.8; take the average of $\bar{\omega }$_1_ and $\bar{\omega }$_2_, and we finally get *ω* = 0.9222.

Heat dissipation *Q*_2_ of aeration tank, sedimentation tank and circulation tank surface can be expressed as:(18)$$Q_{2}=P_{wtd}\cdot A_{s}\left(0.0174V+0.0229\right)\left(P_{td}-P_{ta}\right)\frac{760}{B}$$

The surfaces of the aeration tank, sedimentation tank and circulation tank also carry out convective heat transfer *Q*_3_ with the air.
(19)$$Q_{3}=h_{ad}A_{s}\left(\bar{t_{d}}-t_{a}\right)$$

The heat lost *Q*_4_ from the bottom of the tank, the wall of the tank, the pipeline and the equipment, etc., should be 20% of the evaporation loss on the surface of the pool water according to experience:(20)$$Q_{4}=0.2Q_{2}$$

The heat loss *Q*_5_ from sewage discharge can be written as:(21)$$Q_{5}=0.1Q_{2}$$

By summing up all the types of heat dissipation mentioned above, the aggregate heat dissipation coefficient *ψ* can be calculated using the following relation:(22)$$Q_{s}=\psi \left(\bar{t_{d}}-t_{a}\right)$$
(23)$$\psi =2\cdot 2260\cdot A_{s}(0.0174V+0.0229)+1.3h_{ad}A_{s}$$

The heat release of flue gas is equal to the total heat gain of flue gas and slurry:(24)$$M_{g}c_{p,g}\left(t_{g\left(in\right)}-t_{g\left(out\right)}\right)=\omega h_{ad}A_{gd}\left(\frac{t_{g\left(in\right)}+t_{g\left(out\right)}}{2}-\bar{t_{d}}\right)$$
(25)$$t_{y\left(out\right)}=\frac{\left(\rho_{g}c_{p,g}V_{g}-\frac{1}{2}\omega h_{ad}A_{gd}\right)}{\left(\rho_{g}V_{g}c_{g}+\frac{1}{2}\omega h_{ad}A_{gd}\right)}t_{y\left(in\right)}+\frac{\omega h_{ad}A_{gd}}{\left(\rho_{g}V_{g}c_{p,g}+\frac{1}{2}\omega h_{ad}A_{gd}\right)}\bar{t_{d}}$$

By reconfiguring these relations and setting initial conditions $\left(\tau =0,\;t_{a}=t_{d0}=20\;{}^{\circ}C,\;A_{s}=100{\;m}^{2}\right)$, the equilibrium slurry temperature can be written as:(26)$$\bar{t_{d}}=t_{d0}+\left(\frac{\omega h_{ad}A_{gd}\cdot \bar{t_{y}}+\psi t_{a}}{\omega h_{ad}A_{gd}+\psi }+t_{d0}\right)\left[1-\exp\left(-\frac{\omega h_{ad}A_{gd}+\psi }{M_{p}c_{p,d}}\tau \right)\right]$$

### 2.5. Model for Flue Gas Desulfurization-Waste Heat Recovery System

The flue gas desulfurization-waste heat recovery system is composed of a waste heat recovery heat exchanger on the basis of the original flue gas desulfurization system in order to ensure that the slurry temperature does not continue to rise during the cycle. The system flow chart is shown in Figure 5. *Q_x_* indicates the heat taken away by the heat exchanger to ensure the slurry temperature.

According to the balance of heat, the following formula can be obtained:(27)$$Q_{1}-Q_{s}-Q_{x}=M_{d}c_{p,d}\frac{d\bar{t_{d}}}{d\tau }$$

Modeling in a similar way to Section 2.4, the slurry temperature can be expressed as:(28)$$t_{d}=t_{d0}+\left(\frac{\omega h_{ad}A_{gd}\cdot \bar{t_{g}}+\psi t_{a}+k_{ed}A_{ed}t_{x}}{\omega h_{ad}A_{gd}+\psi +k_{ed}A_{ed}}+t_{w0}\right)\left[1-\exp\left(-\frac{\omega h_{ad}A_{gd}+\psi +k_{ed}A_{ed}}{M_{p}c_{p}}\tau \right)\right]$$

Based on the above modeling, the heat transfer characteristic parameters and structural parameters of the required heat exchanger are calculated and analyzed. The area of the heat exchanger is adjusted according to the change of various influencing factors to reach the appropriate temperature of the desulfurization slurry. The main parameters of the designed immersion heat exchanger are shown in Table 3.

## 3. Results and Discussion

The slurry flows from top to bottom, and the direction of flue gas is opposite. With the inlet of the slurry as x = 0 and the direction of the slurry flow as the positive axis, the coordinate system was established. The reaction of the slurry with different droplet diameters (0.5 mm, 0.7 mm, 0.9 mm) was studied.

### 3.1. Effect of the Key Parameters in the System on Desulfurization Efficiency

#### 3.1.1. Effect of the Reaction Zone Height on Desulfurization Efficiency

The goal of the desulfurization tower is high-efficiency desulfurization. By analyzing the change of SO_2_ concentration in the flue gas with the height of the reaction zone, the desulfurization situation in the tower can be grasped.

The changes in the content of SO_2_ in the flue gas and the desulfurization efficiency with the height of the reaction section are shown in Figure 6a. It can be found that no matter what the droplet diameter is, the desulfurization efficiency of slurry to SO_2_ decreases with the increase of the reaction zone height, but the decreasing rate increases. This is because, with the progress of the reaction, the slurry continuously absorbs the SO_2_ in the flue gas resulting in a continuous decline in the treatment capacity of the slurry to SO_2_. Comparing the change in the desulfurization efficiency of the slurry to SO_2_ in the case of different diameter droplets with the reaction zone height, it can be seen that the desulfurization efficiency of slurry to SO_2_ is largely affected by the initial diameter of droplets. The smaller the diameter is, the higher the desulfurization efficiency of slurry to SO_2_ is, but the faster the decreasing rate is. In the case of a smaller diameter, the content of sulfur dioxide in the flue gas with the same content of sulfur dioxide can be reduced, which is beneficial to environmental protection. In other words, the particle size, to a large extent, directly determines how much sulfur dioxide is reacted in the flue gas.

#### 3.1.2. Effect of the Flue Gas Inlet Temperature on Desulfurization Efficiency

The outlet temperature of flue gas and the moisture content in the flue gas can indirectly reflect the effect of heat and mass transfer in the desulfurization tower. In fact, because there is no suitable way to recover the energy carried by flue gas, the lower the outlet temperature and moisture content, the better.

Figure 6b show that in the case of different droplet diameters, the desulfurization efficiency changes with the flue gas inlet temperature. Under the same droplet diameter, the desulfurization efficiency fluctuates little with the flue gas inlet temperature. The initial equivalent diameter of droplets has a slight influence on the desulfurization efficiency, and the desulfurization efficiency is the highest when the droplet diameter is 0.5 mm. According to the above results, it can be found that when the flue gas inlet temperature varies from 145 °C to 175 °C, the effect it has on the parameters in the desulfurization towers is not particularly obvious.

#### 3.1.3. Effect of the Slurry Inlet Temperature on Desulfurization Efficiency

The desulfurization of slurry is a continuous cyclic reaction. Under the actual operation, the slurry temperature of the desulfurization tower continues to increase. At a certain temperature, the energy of the slurry can continue to be used. Therefore, to a certain extent, the higher the outlet temperature of the slurry, the better.

Figure 6c describe the correlation between the desulfurization efficiency and the change of slurry inlet temperature in the case of different droplet diameters. The desulfurization efficiency of the tower does not change much before 40 °C, and all of them remain relatively high. With the increase of the slurry inlet temperature, the desulfurization efficiency began to decrease rapidly after 45 °C. When the diameter changes, by comparing the three straight lines, it can be found that the desulfurization efficiency is greatly affected by the initial diameter of the droplet. At the same initial slurry inlet temperature, the smaller the diameter is, the higher the desulfurization efficiency is. It can be found that the desulfurization efficiency is insensitive to the change of slurry temperature before 50 °C, and then it will be greatly affected by the change of slurry temperature.

#### 3.1.4. Effect of the Gas-Liquid Ratio on Desulfurization Efficiency

In the actual project of wet magnesium desulfurization, the liquid–gas ratio is generally between 1 and 3, according to experience. In order to improve the efficiency of desulfurization and ensure that sulfur dioxide is absorbed as much as possible, it is natural that the larger the liquid–gas ratio, the better. However, a large liquid–gas ratio will lead to a serious waste problem, so it is necessary to find a suitable value.

Figure 6d show the relationship between the desulfurization efficiency and the liquid–gas ratio in the case of different droplet diameters. The desulfurization efficiency decelerates slowly with the increase of the liquid–gas ratio. When the diameter is constant, by a single curve taken as an example to analyze, the desulfurization efficiency increases rapidly with the increase of the liquid-gas ratio and then becomes flat. Comparing the three straight lines, it can be found that under the same liquid–gas ratio, the smaller the equivalent diameter, the greater the desulfurization efficiency. However, when the liquid–gas ratio is greater than three, the effect of the liquid–gas ratio on the desulfurization efficiency is small.

To sum up, the equivalent diameter of the slurry droplet directly affects the heat and mass transfer capacity of the slurry to a great extent. In the case of the smaller diameter, the heat and mass transfer are relatively more active. As a result, under the condition of ensuring the stability of the system, the selection of smaller diameters has a better effect on the desulfurization process.

### 3.2. Effect of Working Conditions on the Integrated System

From the analysis of the modeling formula of the system, it is known that the performance of the system is mainly determined by heating and heat dissipation; however, heating and heat dissipation have different influence parameters. Therefore, a control variable method is used to discuss the different influence parameters. According to the simplification, the equation of the slurry temperature with time is obtained. The influence of heat dissipation on the slurry temperature is determined by the control variable method. The main influencing factors of heat dissipation are wind speed, heat dissipation area and ambient temperature by the definition of *ψ*.

#### 3.2.1. Effect of Wind Speed on the Integrated System

Figure 7 show the relationship between slurry equilibrium temperature and wind speed. From the analysis of a single curve, the slurry temperature increases rapidly at first and then tends to a smooth curve after approximately 10 h. Subsequently, the rising trend of the temperature slowed down, and the temperature remained stable after twenty hours. According to the longitudinal comparison, in the case of different wind speeds, the rising speed of the final desulfurization slurry temperature is different, and the stable slurry temperature is also unequal. Generally speaking, the smaller the wind speed, the faster the temperature rises. It can be seen from Figure 7 that the slurry equilibrium temperature decreases with the increase of wind speed. The reason is that the increase of wind speed leads to the increase of evaporation heat dissipation.

#### 3.2.2. Effect of Heat Dissipation Area on the Integrated System

Figure 8 show the relationship between slurry equilibrium temperature and heat dissipation area. According to Figure 8, the slurry equilibrium temperature decreases gradually with the increase of the heat dissipation area of the sedimentation tank and circulation tank in the outdoor aeration tank. The temperature first increased rapidly and then began to flatten at approximately 10 h. After 10 h, the temperature reached a stable level and did not change. Longitudinal comparison analysis shows that the smaller the heat dissipation area of the outdoor aeration tank, the smaller the sedimentation tank and the circulation tank, the faster the temperature rises. In the case of the different surface areas of the heat sink, the time required to increase to the most suitable temperature for desulfurization is different.

#### 3.2.3. Effect of Ambient Temperature on the Integrated System

Figure 9 show the relationship between slurry equilibrium temperature and ambient temperature. It can be found that the slurry equilibrium temperature increases gradually with the increase of ambient temperature. The influence of ambient temperature on slurry temperature with time is similar to that of the above heat dissipation area. The vertical comparison shows that the final desulfurization slurry temperature rises at different speeds under different ambient temperatures. The higher the ambient temperature of the flue gas is, the faster the temperature rises. The time required to increase the equilibrium temperature of the desulfurization slurry is different under the temperature of different environments. The higher the ambient temperature of the flue gas, the faster the temperature rises. The time required to increase the equilibrium temperature of the desulfurization slurry is unequal under the different ambient temperatures. The higher the ambient temperature, the higher the equilibrium temperature of the desulfurization slurry.

### 3.3. Effect of Working Conditions on the Heat Transfer Area

As is shown in Figure 6c, when the temperature of desulfurization slurry increases by 1 °C–2 °C, the efficiency of desulfurization will decrease by 4% and 5%. The change of temperature has an obvious effect on the desulfurization efficiency. When the slurry temperature is higher than 55 °C, the desulfurization efficiency can only reach 60–65% of that under normal conditions, which greatly reduces the working efficiency of the desulfurization tower and causes the waste of the desulfurizer. In order to ensure desulfurization efficiency, it is necessary to keep the final temperature of the slurry at less than 55 °C. Therefore, the desulfurization system needs to add a waste heat exchanger to absorb excess heat.

The influence of multiple factors on the heat transfer area of the waste heat exchanger is analyzed in order to achieve maximum heat transfer efficiency. The heat transfer efficiency can be obtained from the comparison of the heat transfer area when reaching a certain slurry equilibrium temperature. By analyzing the mathematical modeling formula, we can see that its influence factors are the same as those of the integrated system.

#### 3.3.1. Effect of Wind Speed on the Heat Transfer Area

Figure 10 show the relationship between heat transfer area and wind speed. When the slurry equilibrium temperature is constant, the area of the replacement waste heat exchanger is negatively correlated with the wind speed. In a practical sense, the greater the wind speed, the less heat is replaced. When the wind speed is constant, the equilibrium temperature of the desulfurization slurry is positively correlated with the area of the replacement waste heat exchanger. The higher the corresponding slurry equilibrium temperature is, the less the area is needed, and the less the waste heat can be utilized. When the wind speed is higher, the area difference of replacement waste heat exchanger is more obvious. Therefore, under all certain conditions, the smaller the wind speed, the more appropriate. When the wind speed is 0 m/s, it is in the best working condition. The specific operation method is to add a cover to the heat dissipation surface of the aeration tank, sedimentation tank and circulation tank, which can effectively reduce unnecessary heat dissipation waste.

#### 3.3.2. Effect of Heat Dissipation Area on the Heat Transfer Area

The relationship between the heat transfer area and the heat dissipation area of the aeration tank, sedimentation tank and circulating tank can be seen in Figure 11. When the slurry equilibrium temperature is constant, the area of the replacement waste heat exchanger is negatively correlated with the heat dissipation area. The larger the heat dissipation area, the smaller the required heat exchanger area. When the area of the aeration tank is constant, the larger the area of the replacement heater, the lower the slurry equilibrium temperature. Under all conditions, the smaller the surface area of the slurry pool, the higher the equilibrium temperature. However, with the reduction of the heat dissipation area of the aeration tank, the function of the sedimentation tank and the circulation tank cannot be fully reflected, resulting in a great decline in the reaction effect. Therefore, there is an optimal working point, which can not only ensure that the role of the aeration tank sedimentation tank and circulation tank can be fully reflected but also effectively reduce the unnecessary waste of heat dissipation.

#### 3.3.3. Effect of the Lumped Average Temperature of the Flue Gas on the Heat Transfer Area

When the heat dissipation capacity of the aeration tank of the desulfurization tower is certain, the heating capacity of the desulfurization tower determines the amount of energy that the final heat exchanger can exchange. Figure 12 show the relationship between the heat transfer area and the lumped average temperature of the flue gas. The area of the heat exchanger is positively correlated with the lumped average temperature of flue gas. When the lumped average temperature of flue gas is constant, the area of the heat exchanger for replacement waste heat is larger in order to achieve a lower equilibrium temperature of the desulfurization slurry. The smaller the lumped average temperature of flue gas, the more obvious the difference of the heat exchanger area of replacement waste heat. With the increase of the lumped average temperature of the flue gas, it means that the flue gas temperature in and out of the desulfurization tower will increase, which will increase the heat that can be raised at the heat exchanger, but the cooling effect of the desulfurization tower on the flue gas is not so obvious. If the temperature of the desulfurization flue gas is too high, it will damage the equipment to a certain extent. Therefore, there is a point of optimal working conditions.

#### 3.3.4. Effect of Ambient Temperature on the Heat Transfer Area

Figure 13 show the relationship between the area of the waste heat exchanger and outdoor ambient temperature. When the slurry equilibrium temperature is constant, the heat exchanger area of replacement waste heat is positively correlated with the temperature of the environment. The higher the ambient temperature, the more heat needs to be replaced in the same case. When the ambient temperature is constant, the lower the equilibrium temperature of the final desulfurization slurry required and more heat transfer area is needed. Generally speaking, the ambient temperature is uncontrollable, but we can do some thermal insulation measures, such as setting up the wall of the thermal insulation pool, adding a lid to the surface of the open slurry pool and so on.

### 3.4. Economic Feasibility Analysis

According to the economic benefit, the whole desulfurization tower and the heat recovery system are studied and analyzed. Under the conditions of the construction costs of the desulfurization tower and waste heat recovery device, the operating raw material cost of the system, the expenses of operation and maintenance, the thermal process parameters, the environmental parameters and the recoverable heat energy are known, the effects of uncontrollable conditions such as flue gas inlet temperature and slurry inlet temperature of the system and controllable conditions such as the liquid–gas ratio and wind speed on the surface of open aeration tank on the economic index of the integrated system are studied. At the same time, the maximum value of the economic benefit is taken as the objective function to find the optimal solution. Taking the annual comprehensive benefit index of the integrated system as the objective function, in the process of analysis, the annual comprehensive benefit is converted into the difference between the annual comprehensive income and the comprehensive cost. Taking the working condition (${A=100\;m}^{2},\;v=0\;m/s,\;t_{g}=165\;{}^{\circ}C,\;t_{a}=20\;{}^{\circ}C$) as an example, we can get the results shown in Table 4 and Table 5.

Table 4 and Table 5 show that even if there is an initial investment, the recovery of waste heat is a single increase. Using the calculation of costs and benefits, it can be determined that the cost can be recovered in two years. From an economic point of view, it is a suitable investment project.

## 4. Conclusions

One new type of flue gas desulfurization-waste heat recovery system was put forward. A model for heat and mass transfer in the spray desulfurization tower and a desulfurization tower and open slurry pool integrated system were set up. The effects of the reaction zone height, flue gas inlet temperature, slurry inlet temperature and gas–liquid ratio on desulfurization efficiency were discussed. The influence of working conditions on the working characteristics of the system were studied. The main results and conclusions are summarized as follows:(1)The mathematical and physical models were established from the liquid phase and gas phase, respectively. Through these models, five related models about *Q*, *x_g_*, *t_d_*, *t_g_* and $c_{so_{2}}$ were determined. The numerical method was developed here to ensure the correctness and convergence of the results.(2)The desulfurization efficiency was analyzed thoroughly. The desulfurization efficiency decreases gradually along the height of the reaction zone. The smaller the initial equivalent diameter of the droplet, the greater the desulfurization efficiency. The desulfurization efficiency tends to be flat when the inlet temperature of flue gas is between 155 °C and 175 °C. When the slurry inlet temperature is lower than 40 °C, the desulfurization efficiency does not change much with the slurry temperature. When the slurry inlet temperature is higher than 55 °C, the desulfurization efficiency is highly affected by the slurry temperature change.(3)The influence of the working conditions on the desulfurization system was also discussed in detail. It can be found that the order of influence degree to the desulfurization system is as follows: the outdoor wind speed is the strongest, the surface area of the open slurry pool is the second, the average flue gas temperature is slightly weaker and the outdoor ambient temperature is the weakest.(4)The economic analysis of the desulfurization system was carried out using the annual cost method, and the comprehensive net benefit of the known year is positive, which proves the economic feasibility of the desulfurization system.

## Figures and Tables

**Figure 1 membranes-12-00047-f001:**
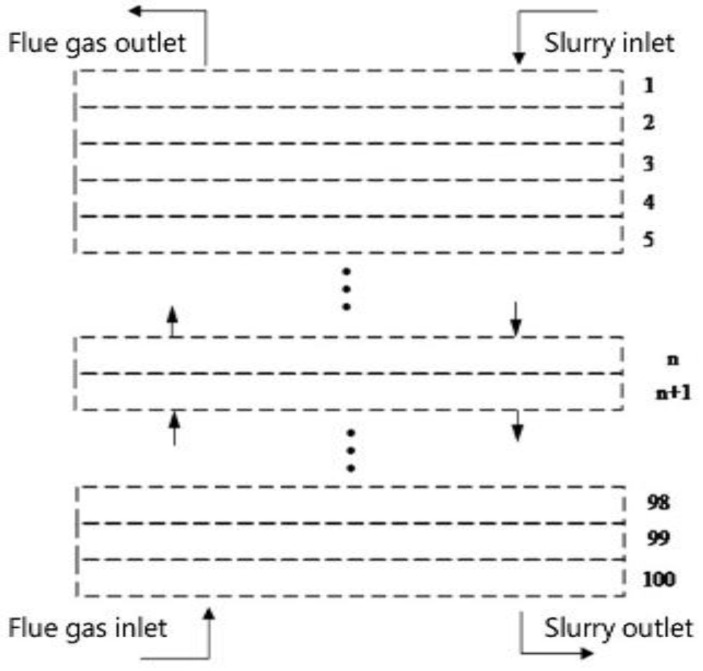
Numerical calculation schematic diagram of desulfurization tower.

**Figure 2 membranes-12-00047-f002:**
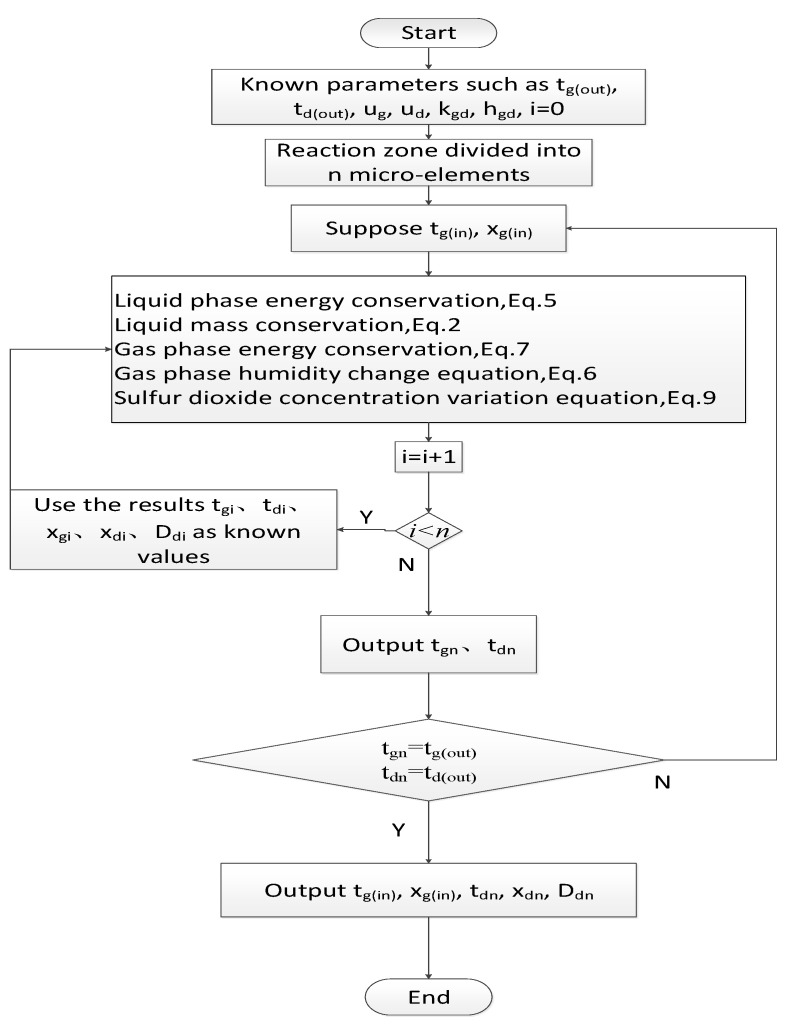
Numerical calculation flow chart of desulfurization tower.

**Figure 3 membranes-12-00047-f003:**
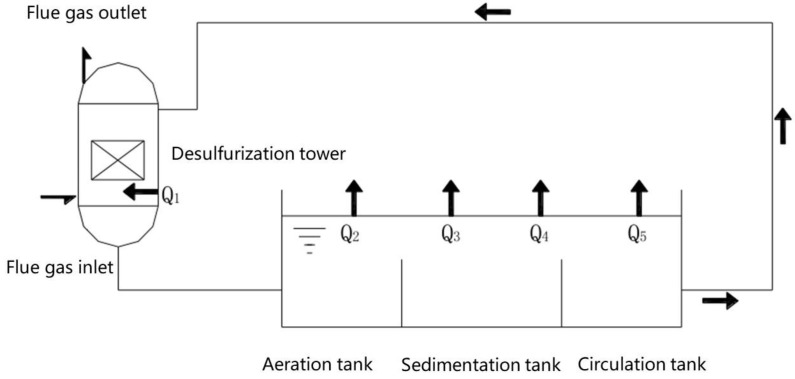
Flue gas desulfurization system diagram of a coal-fired boiler.

**Figure 4 membranes-12-00047-f004:**
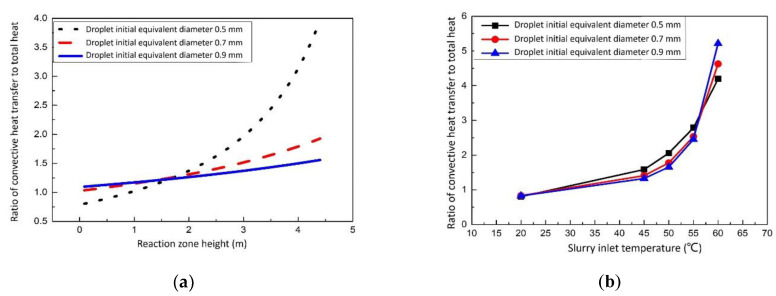
Ratio of convective heat transfer to total heat of flue gas and slurry. (**a**) Effect of reaction zone height on the ratio of convective heat transfer to total heat. (**b**) Effect of slurry inlet temperature on the ratio of convective heat transfer to total heat.

**Figure 5 membranes-12-00047-f005:**
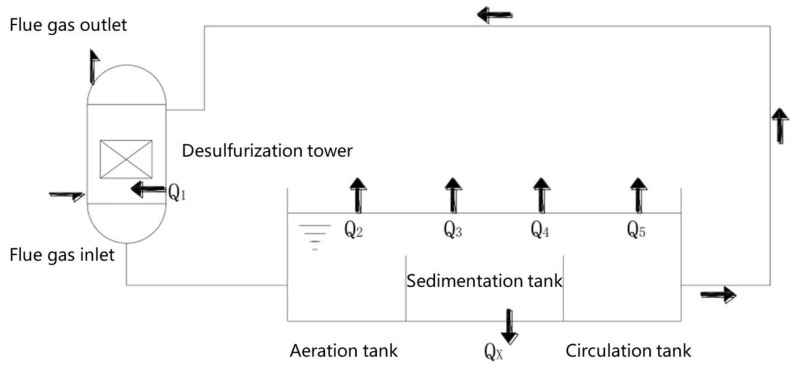
Flow chart of flue gas desulfurization and waste heat recovery system.

**Figure 6 membranes-12-00047-f006:**
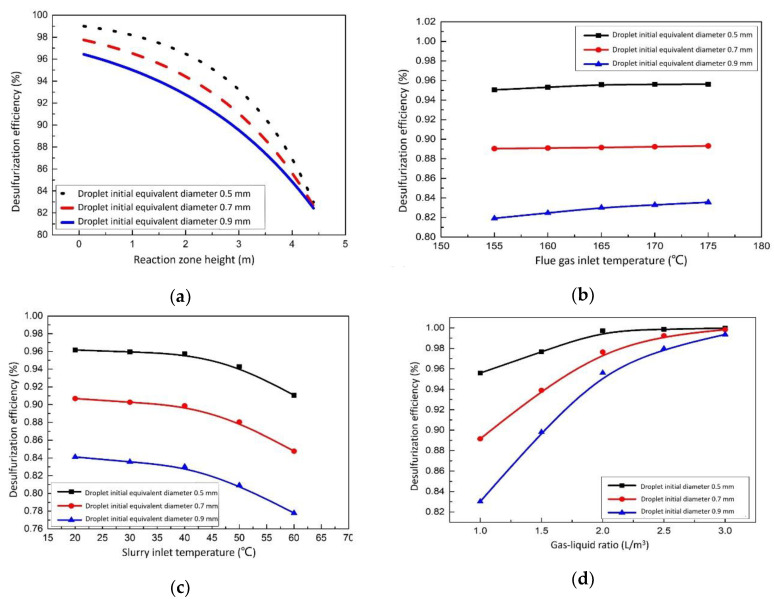
Effect of different parameters on desulfurization efficiency. (**a**) Effect of the reaction zone height on desulfurization efficiency. (**b**) Effect of the flue gas inlet temperature on desulfurization efficiency. (**c**) Effect of the slurry inlet temperature on desulfurization efficiency. (**d**) Effect of the gas–liquid ratio on desulfurization efficiency.

**Figure 7 membranes-12-00047-f007:**
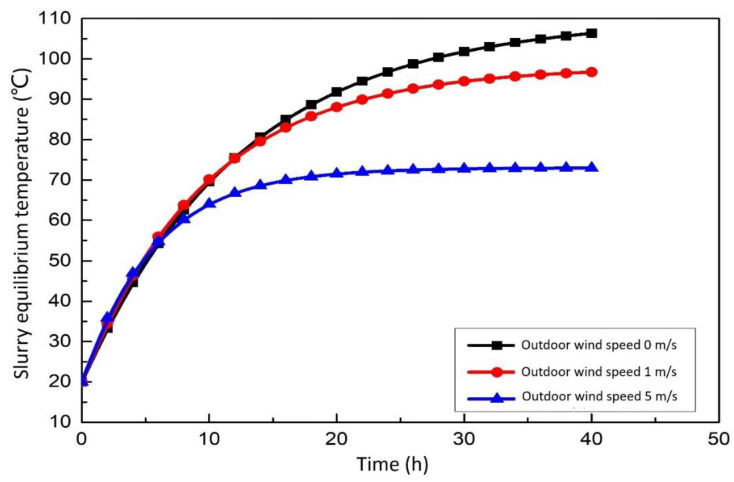
Effect of wind speed on the slurry equilibrium temperature.

**Figure 8 membranes-12-00047-f008:**
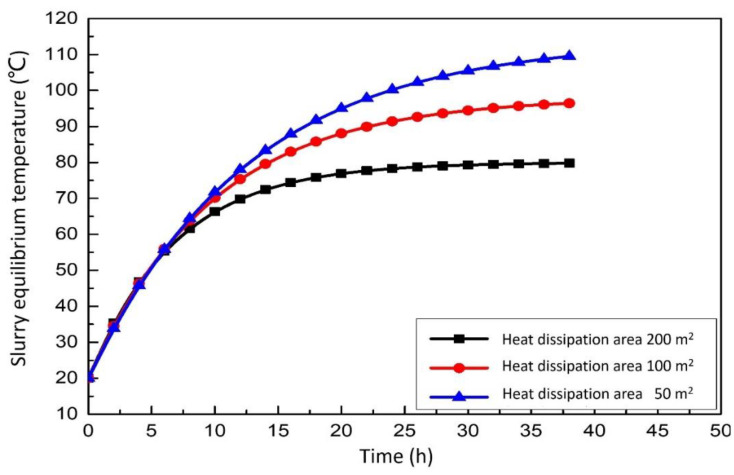
Effect of heat dissipation area on the slurry equilibrium temperature.

**Figure 9 membranes-12-00047-f009:**
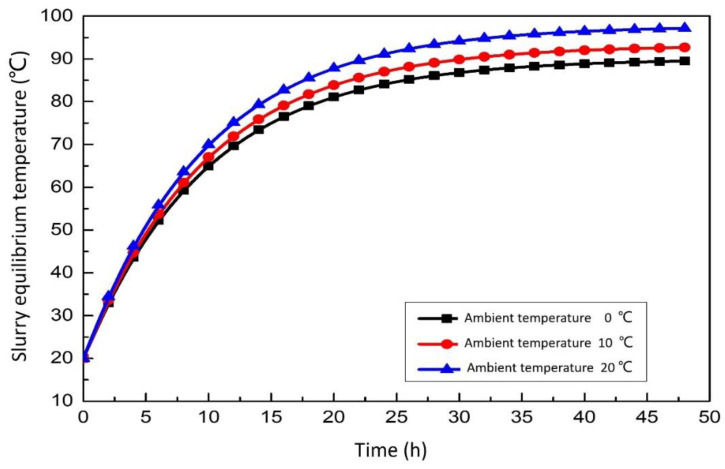
Effect of ambient temperature on the slurry equilibrium temperature.

**Figure 10 membranes-12-00047-f010:**
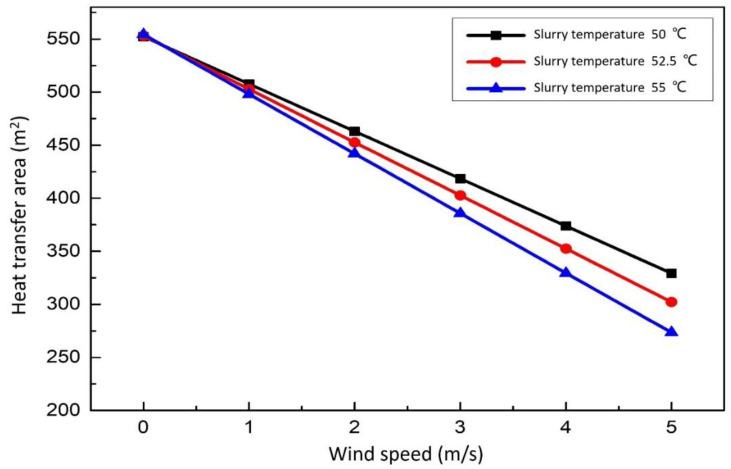
Effect of wind speed on the heat transfer area.

**Figure 11 membranes-12-00047-f011:**
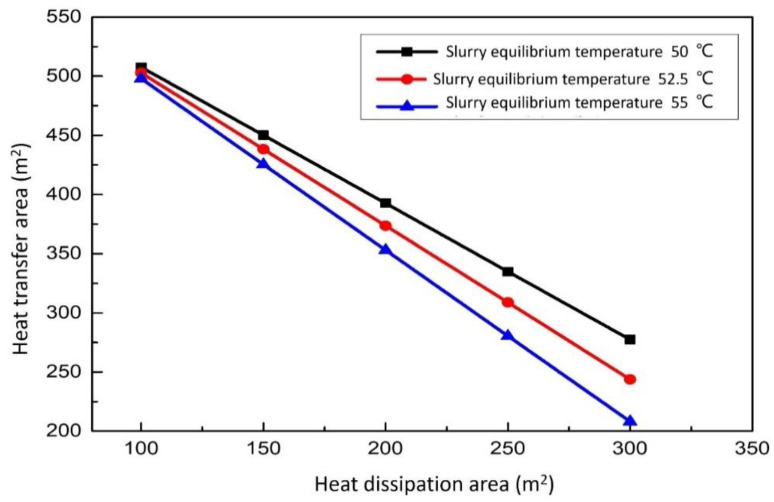
Effect of heat dissipation area on the heat transfer area.

**Figure 12 membranes-12-00047-f012:**
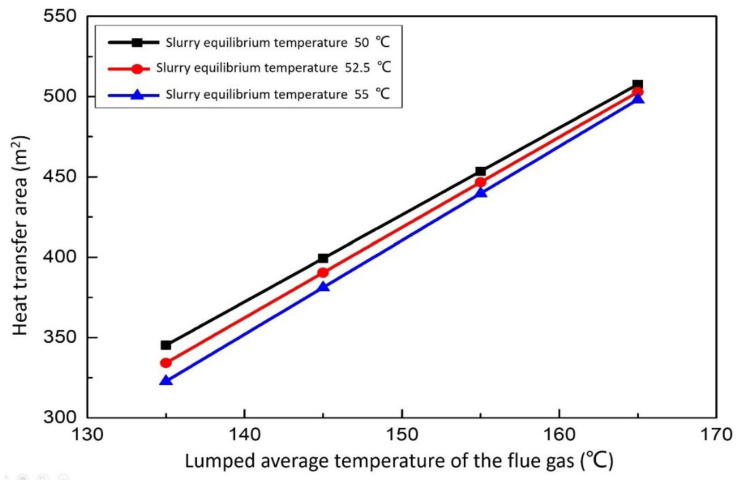
Effect of the lumped average temperature of the flue gas on the heat transfer area.

**Figure 13 membranes-12-00047-f013:**
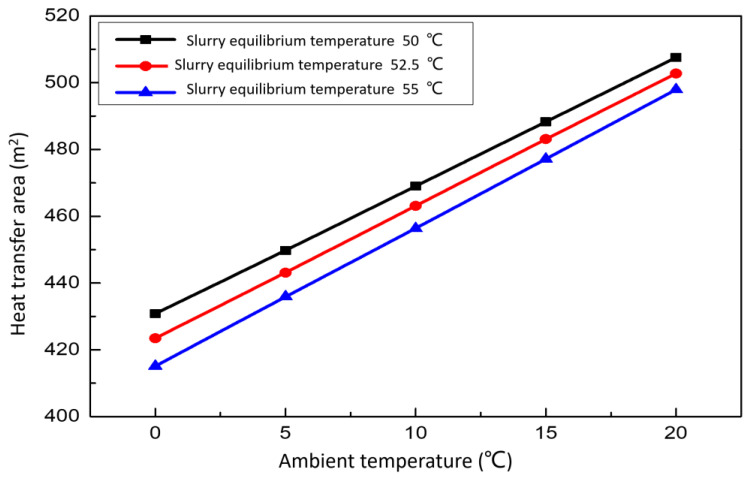
Effect of ambient temperature on the heat transfer area.

**Table 1 membranes-12-00047-t001:** The working media and structural parameters of the desulfurization tower.

Item	Value
Boiler parameters (t/h)	50
Coal burning quantity (t/h)	8
Flue gas volume (Nm^3^/h)	110,000
Desulfurizer concentration (%)	0.15
Desulfurizer density (kg/m^3^)	1100
Sulfur dioxide mass concentration (mg/m^3^)	827
Flue gas moisture content (kg/kg)	0.05
Mg–S ratio	3.3
Liquid–gas ratio	1.67
Air excess factor	1.6
Flue gas velocity (m/s)	4
Flue gas temperature (°C)	165
Droplet velocity (m/s)	5
Droplet temperature (°C)	45
Reaction zone length (m)	4.5

**Table 2 membranes-12-00047-t002:** The ratio of convective heat transfer to total heat transfer.

Item	Value				
Reaction zone height (m)	0	1	2	3	4
$\omega_{1}$	1	1.0222	1.0444	1.0666	1.0888
$t_{d}\;({}^{\circ}C)$	30	35	40	45	50
$\omega_{2}$	1.0024	0.87666	0.75176	0.62554	0.498

**Table 3 membranes-12-00047-t003:** The main parameters of the designed immersion heat exchanger.

Item	Flue Gas	Secondary Water
Inlet temperature (°C)	165	45
Outlet temperature (°C)	121	50
Flow (m^3^/h)	110,925	200
Velocity of flow (m/s)	-	1.81
Pipe diameter	-	DN200

**Table 4 membranes-12-00047-t004:** The cost parameters of flue gas desulfurization-waste heat recovery system.

Item	Value
Heat exchanger (Ten thousand yuan/m^2^)	0.15
Heat exchanger area (m^2^)	550
Instrument pump pipeline (Ten thousand yuan)	21
Civil engineering installation (Ten thousand yuan)	25
Cover plate (Ten thousand yuan)	4
Tax revenue (Ten thousand yuan)	15.63
Total (Ten thousand yuan)	148.13

**Table 5 membranes-12-00047-t005:** The benefit parameters of flue gas desulfurization-waste heat recovery system.

Item	Value
Operating time (h/year)	4380
Conversion price of calorific value (yuan/GJ)	40
Heat recovery efficiency (%)	80
Saving energy (kW)	1461
Total (Ten thousand yuan)	73.7

## Data Availability

Not applicable.

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
