# Peer review of "Coal-Fired Boiler Flue Gas Desulfurization System Based on Slurry Waste Heat Recovery in Severe Cold Areas"

_membranes, 2021, doi:10.3390/membranes12010047_

Round 1

Reviewer 1 Report

Dear authors,

Comments are given in PDF file. Please check.

Reviewer 2 Report

I send a pdf with the comments and suggestions for authors.

Reviewer 3 Report

Comments and Suggestions for Authors

Dear Authors,

The Title:

Coal-fired boiler flue gas desulfurization system based on 2 slurry waste heat recovery in severe cold areas

I have to read your manuscript with great attention and interest. The material is consistent, comprehensive and complete.

The authors of the manuscript presented a theoretical model with a computational method for the wet flue gas desulphurization process. The cooperation system of the spray tower, open slurry pool and desulphurization system with heat recovery was developed.

The submission falls within the scope of the journal and is sufficiently original, and I have a remark, so I recommended the publication after MAJOR REVISIONS.

    1. Please explain all symbols and give explanations before Introduction  
    2. Read your work carefully as errors appear: see line 147, line 172, line 197, line 200, line 268, line 281, line 298, line 324, line 332, line 338, 339, 351, 365, etc.
  • improve the quality of the Fig. 4., Fig. 6., 7, 9, 10, 11, 12,13
  • Work out conclusions in the form of short points

Round 2

Reviewer 3 Report

The authors referred to all comments and recommendations. I propose to accept the article for publication.